# MOTION-AGENT: A CONVERSATIONAL FRAMEWORK FOR HUMAN MOTION GENERATION WITH LLMS

**Qi Wu**[1]*, **Yubo Zhao**[1]*, **Yifan Wang**[1], **Xinhang Liu**[1], **Yu-Wing Tai**[2], **Chi-Keung Tang**[1]
[1]The Hong Kong University of Science and Technology
[2] Dartmouth College

## ABSTRACT

While previous approaches to 3D human motion generation have achieved notable success, they often rely on extensive training and are limited to specific tasks. To address these challenges, we introduce **Motion-Agent**, an efficient conversational framework designed for general human motion generation, editing, and understanding. Motion-Agent employs an open-source pre-trained language model to develop a generative agent, **MotionLLM**, that bridges the gap between motion and text. This is accomplished by encoding and quantizing motions into discrete tokens that align with the language model's vocabulary. With only 1–3% of the model's parameters fine-tuned using adapters, Motion-LLM delivers performance on par with diffusion models and other transformer-based methods trained from scratch. By integrating MotionLLM with GPT-4 without additional training, Motion-Agent is able to generate highly complex motion sequences through multi-turn conversations, a capability that previous models have struggled to achieve. Motion-Agent supports a wide range of motion-language tasks, offering versatile capabilities for generating and customizing human motion through interactive conversational exchanges. Project page: https://knoxzhao.github.io/Motion-Agent

## 1 INTRODUCTION

Large Language Models (LLMs) have recently attracted much attention in both industry and academia. Many LLMs, such as GPT-4 (Achiam et al., 2023), LLaMA (Touvron et al., 2023), Gemma (Team et al., 2024a), have shown their advanced capabilities, robustness and generalization across various downstream tasks. These progresses have motivated researchers to explore the application of LLMs in multimodal tasks, integrating them with modalities such as images (Koh et al., 2024), videos Zhang et al. (2023a), audio (Borsos et al., 2023; Huang et al., 2023), and more, resulting in promising outcomes in understanding these different modalities. However, the utilization of LLMs in the context of multimodal *generation*, particularly of 3D human motion, remains underexplored, which is crucial for advancing robots and humanoid applications.

Research in 3D human motion has explored various language-related tasks, including text-conditioned motion generation (Zhang et al., 2023b; Guo et al., 2022a; Tevet et al., 2023; Zhang et al., 2022; Shafir et al., 2024; Guo et al., 2024; Jiang et al., 2024b), motion captioning (Guo et al., 2022b; Jiang et al., 2024b), motion reasoning (Endo et al., 2023; Jiang et al., 2024c). However, existing methods often require extensive training, leading to high computational demands and inefficiency. These models are typically trained on task-specific data, making them data-dependent and limiting their ability to generalize across diverse scenarios. They also struggle with handling long, complex prompts with performance degradation. Furthermore, most existing models lack the capability to support multi-turn conversational interactions, thus limiting both the generation and refinement processes, and restricting the ability to create dynamic, interactive systems that can seamlessly generate and allow editing motions through dialogue.

Moving forward with the most recent LLM and MLLM development, in this work, we propose **Motion-Agent**, a multimodal framework that leverages the generalization and flexibility of pre-

---

*Equal contribution.

`User$` What is this person doing in the gym? What is the type of the exercise and what are the training goals?

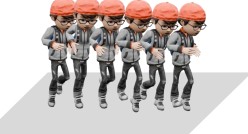

`Motion-Agent$` The person is running on a treadmill. Running on a treadmill is a type of aerobic exercise that primarily focuses on cardiovascular fitness. The training goal can vary but often includes improving heart health, increasing stamina and endurance, losing weight, or preparing for competitive running events.

`User$` Can you generate a motion containing an example of both an aerobic exercise and an anaerobic exercise?
`Motion-Agent$`

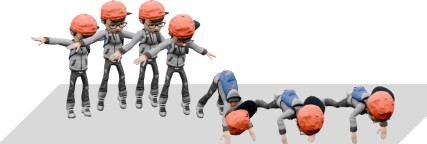

`User$` Explain the motion you have just generated in detail.
`Motion-Agent$` The motion sequence generated involves two exercises. First, the person performs jumping jacks, which is a physical jumping exercise where the body's limbs are moved repetitively in and out while the hands clap overhead, then quickly return to the sides. This exercise is aerobic, helping to increase heart rate and improve cardiovascular fitness. Second, the person does push-ups, which involve repeatedly raising and lowering the body using the arms while lying face down. This exercise is anaerobic, focusing on strengthening the upper body and core muscles by resisting body weight.

*cont'd*
`User$` Well, I don't want the person to do push-ups now, can you make the person do squats instead?
`Motion-Agent$`

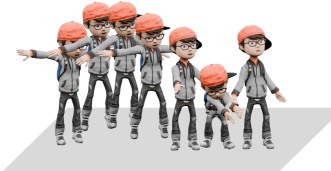

`User$` After the exercise, the person is tired and proceeds to sit down to rest.
`Motion-Agent$`

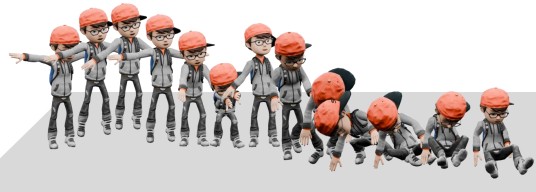

`User$` Replace squats with running in place, and I want the person to sit on a chair in front instead of on the ground.
`Motion-Agent$`

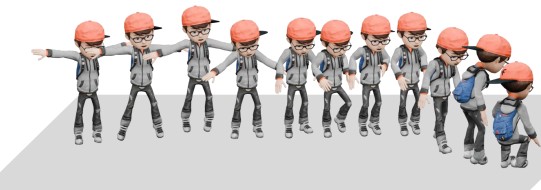

Figure 1: **Multi-turn Conversation Between User and Motion-Agent**. First Turn: Motion Understanding; Second Turn: Motion Generation; Third Turn: Motion Understanding with Previously Generated Motion; Fourth Turn: Motion Editing; Fifth Turn: Continue Motion Generation; Last Turn: Motion Editing on Long Sequence. Note that all turns are continuous.

trained LLMs. Central to the framework is our new generative agent, **MotionLLM**, the incorporation of which eliminates the need for extensive pre-training by employing lightweight adapter-based fine-tuning of a pre-trained LLM. Unlike MotionChain (Jiang et al., 2024c), which requires pre-training and large datasets for extensive instruction tuning to achieve conversational control, Motion-Agent integrates MotionLLM with GPT-4 and leverages the LLM's inherent conversational capabilities without additional training. This enables efficient, customizable motion generation, understanding, and multi-turn editing across various tasks.

In Motion-Agent, we first train a pair of motion tokenizer and detokenizer. The motion tokenizer encodes motions into motion embeddings and quantizes them into a set of discrete LLM-understandable tokens using a codebook, while the detokenizer reconstructs tokens back to their original continuous forms. This tokenizer-detokenizer pair enables the translation between continuous motion sequences and discrete tokens, facilitating interaction with the LLM while still allowing for the recovery of the original motions from the tokens. MotionLLM is trained by enriching a pre-trained LLM's vocabulary with these additional motion tokens, while keeping the original text tokens unchanged. Given that motions can be represented as temporal sequences, our tokenization process converts motions into token sequences akin to sentences in natural language. MotionLLM translates between text token sequences and motion token sequences. On top of this, GPT-4 acts as a coordinator, decomposing user instructions to determine the number of calls to MotionLLM and how to structure those calls effectively. The resulting motion token sequences from multiple calls are concatenated and decoded by the detokenizer to produce the final output.

Our Motion-Agent framework leverages pre-trained LLMs in two key ways: (1) fine-tuning a lightweight LLM via adapters to serve as a text-motion translation agent, and (2) using an LLM for conversational interactions without training, thus facilitating multi-turn dialogue for refining

generated motions and producing extended motions by iteratively generating and concatenating sequences. Despite training only a small number of parameters, MotionLLM can achieve competitive results in motion generation (text to motion) compared to those trained-from-scratch models with specialized architectures. In motion captioning (motion to text), MotionLLM achieves state-of-the-art performances, generating semantically accurate and contextually appropriate text descriptions. MotionLLM enables bidirectional translation between text and motion, outperforming other autoregressive models while using fewer trainable parameters, making it an ideal fit for the overall Motion-Agent framework. By combining MotionLLM with GPT-4, Motion-Agent enables versatile dialogue-based motion generation and reasoning, without requiring specific datasets or extra training for these tasks.

To summarize, our contributions include:

- We introduce a simple, efficient conversational framework, Motion-Agent, that utilizes pretrained LLMs and produces strong results in various motion-language tasks.

- We demonstrate the flexibility and versatility of our method by achieving highly customizable motion-language tasks, including long and complex motion generation, multi-turn editing, and multi-turn reasoning.

## 2 RELATED WORK

**Multimodal LLMs** Recent advancements have integrated large language models (LLMs) with multiple modalities such as image, video, music, audio, and point cloud using different approaches (Liu et al., 2024; Han et al., 2024; Wu et al., 2023b; Chen et al., 2023a; Gao et al., 2023). Various approaches have been proposed to align different modalities. For instance, Video-LLaMA (Zhang et al., 2023a) leverages Q-formers to bridge the gap between modalities. PointLLM (Xu et al., 2023b) utilizes a projector to align the feature space of point clouds with the feature space of the LLM. VALLE-X (Zhang et al., 2023d) and LlamaGen (Sun et al., 2024) tokenize inputs from various modalities to connect them with language. On the other hand, emerging research (Wu et al., 2023a; Lu et al., 2024; Du & Kaelbling, 2024; Liu et al., 2025) demonstrates promising results with compositional language models. These models, often composed of smaller specialized components, excel in data efficiency and perform well on unseen distributions, aligning with the design of our framework.

**3D Human Motion Synthesis** Modern works can generate human motions based on a variety of inputs such as action labels (Petrovich et al., 2021; Lee et al., 2023; Guo et al., 2020; Xu et al., 2023a), textual descriptions (Jiang et al., 2024b; Wang et al., 2023; Zhang et al., 2023b; Guo et al., 2022b; Zhou et al., 2023; Tevet et al., 2023; 2022; Guo et al., 2024; Zhang et al., 2022; Dabral et al., 2023; Petrovich et al., 2022; Zhang et al., 2023c; Pinyoanuntapong et al., 2024), control signals (Xie et al., 2024; Wan et al., 2023; Petrovich et al., 2024; Huang et al., 2024; Goel et al., 2023), music or audio (Dabral et al., 2023; Tseng et al., 2022; Zhou & Wang, 2023; Siyao et al., 2022; 2023), and others (Zhong et al., 2024). Particularly, text-guided 3D motion generation or text-to-motion has garnered significant interest. Notably, some diffusion models have emerged as powerful tools, such as Tevet et al. (2023); Shafir et al. (2024); Wang et al. (2023); Zhou et al. (2023); Xie et al. (2024); Zhang et al. (2022). Despite the proficiency in generating motions, diffusion models necessitate manual length control of the generated motions with limited flexibility. In addition to diffusion models, which employ continuous motion representation, discrete token-based methods utilizing Vector Quantized Variational Autoencoders (VQ-VAEs) have also demonstrated promising results. Notable examples include TM2T (Guo et al., 2022b), T2M-GPT (Zhang et al., 2023b), MotionGPT (Jiang et al., 2024b) and MoMask (Guo et al., 2024). Most existing works in both approaches focus on conditional generation to translate between modalities. In our work, we emphasize generating human motion through complex, customized user conversations while proposing a training-efficient approach to bridge these modalities using pre-trained LLMs.

**Conversational Control For Human Motion** Generating 3D human motion through conversation is more flexible, which allows users to customize versatile requests and control motion via iterative refinement. While models like MotionGPT (Jiang et al., 2024b) handle some simple single-turn tasks using instruction tuning, and MotionChain (Jiang et al., 2024c) supports multi-turn interactions by sampling single-turn data into multi-turn training data, both methods rely heavily on extensive

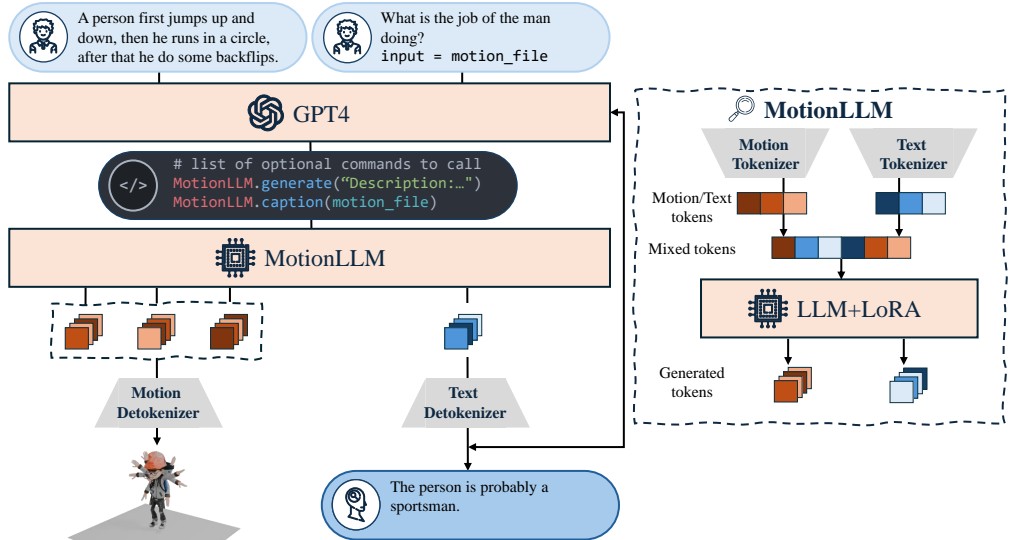

Figure 2: **Motion-Agent pipeline**. GPT-4 can interact with the translation agent (i.e., MotionLLM) to generate or interpret motions based on input requirements. The generated motion tokens are concatenated and decoded, and the textual caption produced by MotionLLM is returned and processed by GPT-4.

instruction tuning and additional data. In contrast, our Motion-Agent framework uses a composition of LLMs to eliminate extra training. By training the translation agent, MotionLLM, solely on the original text-motion paired data, our method eliminates the need for further data or training, resulting in higher efficiency and broader generalizability.

## 3 METHOD

As shown in Fig. 2, our Motion-Agent framework primarily consists of three components: an LLM (i.e., GPT-4) for conversational interaction and prompting control, a pair of motion tokenizer/detokenizer, and a translation agent (i.e., MotionLLM). The text tokenizer is inherited from the LLMs and remains unchanged, while the motion tokenizer and detokenizer are trained together to ensure proper reconstruction of motion sequences. Once trained, the motion tokenizer and detokenizer are kept fixed. The motion detokenizer plays a key role in smoothing the transitions between different motion sequences, ensuring seamless integration of motion outputs.

To ensure bidirectional understanding, our framework also enables motion comprehension. Thus, the agent should also be capable of generating textual captions from given motions upon request. This bidirectional translation is crucial for applications such as answering questions about motions or generating descriptions, where models that can only perform motion generation are not suitable. On the other hand, our proposed MotionLLM can indeed be a good fit, ensuring bidirectional translation within a unified architecture.

### 3.1 THE MOTION-AGENT FRAMEWORK

In this framework, GPT-4 serves as the coordinator of both motion generation and comprehension, enabling seamless interaction between users and a multimodal text-motion agent. The agent is responsible for translating between text and motion modalities. Within the conversation, the input to GPT-4 consists of two components: a fixed instruction prompt $p$, which provides guidelines for interacting with the text-motion agent, and a customized request $c$ from the user. Based on this input, GPT-4 generates a structured plan, determining whether the agent should perform tasks such as motion generation or captioning. It also decides how many times to invoke the agent and specifies the arguments for each invocation. This plan, formatted as a JSON file, is then parsed and executed by the agent to carry out the specified tasks.

For generation, the agent generates motion token sequences corresponding to each set of arguments, and these sequences are concatenated for universal decoding. Specifically, let $G$ represent the agent and $[\mathbf{a}_i]_{i=1}^N$ the arguments for each of the $N$ calls determined by GPT-4. The resulting motion token sequences $\mathbf{z}_i = G(\mathbf{a}_i)$ are concatenated to form a single sequence $\mathbf{z} = (\mathbf{z}_1, \mathbf{z}_2, \ldots, \mathbf{z}_N)$. This sequence is decoded by the decoder $D$ to produce the final motion, $\mathbf{m} = D(\mathbf{z})$, as will be outlined in the tokenization section.

For motion understanding and reasoning, the agent generates textual captions of the motions, which are then returned to GPT-4. This allows GPT-4 to interpret the motion and respond to user queries accordingly, enabling seamless interactions between users and the system through both motion generation and comprehension.

Since LLMs such as GPT-4 possess strong multi-turn conversational abilities, users can continuously ask the model to refine, edit, or extend previous generations, as well as pose additional questions. In response, GPT-4 will re-generate the plan or provide answers, thus providing an interactive and adaptive system. This dynamic interaction leads to a unified framework that supports an exceptionally wide variety of combinations, lengths, and task complexities, offering enhanced flexibility and customization across both motion generation and comprehension.

## 3.2 Motion Tokenization

In order to align better with LLM's next-token prediction mechanism, we tokenize motions into discrete representations using Vector Quantization (VQ) and Variation AutoEncoders (VAE). This VQ-VAE approach is widely adopted by Guo et al. (2022b), Siyao et al. (2022), Siyao et al. (2023) Zhang et al. (2023b), Jiang et al. (2024b), and Guo et al. (2024).

In our motion tokenization, a motion sequence is represented as $\mathbf{m}_{1:T} \in \mathbb{R}^{T \times D}$ and is first encoded using an encoder $E$ to motion embeddings $\mathbf{z}_{1:T/N} \in \mathbb{R}^{T/N \times d}$, where $N$ is the downsampling rate and $d$ is the number of the hidden dimensions. Then the motion embeddings are quantized by a quantizer using a codebook $\mathbf{C} = \{\mathbf{c}_k\}_1^K$, where $K$ is the codebook size and each $\mathbf{c}_k \in \mathbb{R}^d$. The quantization results can be represented as $\hat{\mathbf{z}}_{1:T/N}$, where

$$\hat{\mathbf{z}}_t = \arg\min_{\mathbf{c}_k \in \mathbf{C}} ||\mathbf{z}_t - \mathbf{c}_k||_2$$

The original sequence can be reconstructed by the decoder $D$: $\hat{\mathbf{m}}_{1:T} = D(\hat{\mathbf{z}}_{1:T/N})$.

We follow Zhang et al. (2023b) to optimize the VQ-VAE, using reconstruction loss together with a commitment loss. We also add an additional regularization on the joint positions $\mathbf{p}$ to enhance the generation performance. The loss can be formulated as:

$$\mathcal{L}_{vq} = \underbrace{||\mathbf{m} - \hat{\mathbf{m}}||_1}_{\mathcal{L}_{re}} + \alpha \underbrace{||\mathbf{p} - \hat{\mathbf{p}}||_1}_{\mathcal{L}_p} + \beta \underbrace{||\mathbf{z} - sg[\hat{\mathbf{z}}]||_2}_{\mathcal{L}_{commit}}$$

where $sg[\cdot]$ is the stop-gradient operation, $\alpha$ and $\beta$ are weighting factors. The codebooks are trained using exponential moving average (EMA) and codebooks reset following T2M-GPT (Zhang et al., 2023b). After training, the tokenizers are frozen for further usage.

## 3.3 LLM-based Motion-Language Agent

Following tokenization, the motion representation is discretized into $K$ distinct motion tokens. We utilize the indices of these motion tokens from the codebook to construct the motion token vocabulary $\mathbf{V}_m = \{< \texttt{Motion\_i} >\}_{i=1}^K$. In addition, we introduce special tokens "<$\texttt{Motion}$>" and "<$\texttt{/Motion}$>" to denote the start and end of a motion token sequence. These special tokens, together with the motion tokens, form a new vocabulary set $\mathbf{V}_M$ of size $K + 2$. This vocabulary will then be appended to the pre-trained LLM's vocabulary.

After expanding the LLM's vocabulary, a motion can now be denoted as a token sequence that is understandable by the LLM. During the generation process, the LLM predicts the succeeding token by maximizing the probability $p_\theta(x_t | x_{<t}, c)$, where $x_{1:T}$ is the target token sequence and $c$ represents the prompt. This prediction is performed iteratively in an autoregressive manner. Consequently, the training objective aims to maximize the log-likelihood $\mathcal{L}_{LLM} = -\sum \log p_\theta(x_t | x_{x_{<t}}, c)$.

| Methods | Motion Generation | Captioning | Multi-turn Editing | Reasoning | Composition |
|---|---|---|---|---|---|
| MotionGPT (Jiang et al., 2024b) | short | ✓ | ✗ | ✗ | ✗ |
| MoMask (Guo et al., 2024) | short | ✗ | ✗ | ✗ | ✗ |
| *MotionChain (Jiang et al., 2024c)* | short | ✓ | ✓ | ✓ | ✓ |
| Ours | long | ✓ | ✓ | ✓ | ✓ |

Table 1: Comparison on functionalities among recent motion generation models. Italicized *model* indicates the corresponding model requires pre-training and task-specific tuning.

During the inference process, our approach utilizes instructive prompts such as "Generate a motion that matches the following input human motion description." accompanied by a sentence describing the desired motion. The LLM then proceeds to predict tokens autoregressively until it predicts the "</Motion>" token, indicating the completion of the motion generation. This autoregressive process allows for the generation of motions with variable lengths, adapting to the specific requirements of the given description.

To fine-tune the LLM, we employ LoRA (Hu et al., 2021). Throughout the whole training process, the tokenizer, the embeddings, and the output layer of the original text tokens remain unchanged and frozen. Only the additional adapters are trained. These LoRA adapters are trained for the task at hand (generation or captioning) while maintaining a general architecture where multiple adapters can coexist harmoniously. This approach allows us to leverage the power of LLMs while tailoring them to specific motion-language tasks, ensuring efficient and effective training without altering the core components of the LLM.

## 4 EXPERIMENTS

We assess our Motion-Agent framework with general and complex conversational user inputs, demonstrating its ability to handle intricate, multi-turn interactions. We also evaluate MotionLLM on single-turn motion generation and motion captioning tasks.

### 4.1 EXPERIMENT SETUP

**Datasets.** Our experiments on MotionLLM are conducted with KIT Motion Language Dataset (KIT-ML) (Plappert et al., 2016), HumanML3D (Guo et al., 2022a). KIT-ML contains 3,911 human motion sequences, while HumanML3D dataset, obtained from AMASS (Mahmood et al., 2019) and HumanAct12 (Guo et al., 2020), contains 14,616 human motions sequences with 44,970 textual descriptions. For Motion-Agent, we use the MotionLLM model which is trained on HumanML3D.

**Evaluation Metric.** For motion generation, we follow T2M (Guo et al., 2022a). Global representations of motion and textual descriptions are first extracted with the pre-trained network in (Guo et al., 2022a) and then measured in the following: 1) Text matching: R-precision (Top-1, Top-2, and Top-3 accuracy) by ranking Euclidean distances between motion and text embeddings, and MM Dist, which measures the average distance between text and generated motion embeddings. 2) Generation diversity: quantifies the variance of generated motions across all descriptions. 3) Motion fidelity: FID assesses the distance between the distribution of real and generated motions, reflecting how closely they match real motion distributions. For motion captioning, we follow TM2T (Guo et al., 2022b) to evaluate the quality of motion captioning by facilitating linguistic metrics from natural language studies, including Bleu (Papineni et al., 2002), Rouge (Lin, 2004), Cider (Vedantam et al., 2015), and Bert Score (Zhang et al., 2020).

**Implementation Details.** We utilize GPT-4 (Achiam et al., 2023) as the conversational LLM in our Motion-Agent framework, which offers enhanced textual control and interaction capabilities. In our tokenizer, we set the downsampling rate $N$ to 4, the hidden dimension $d$ to 512, and the codebook size $K$ to 512. The weighting factors $\alpha$ and $\beta$ for $\mathcal{L}_p$ and $\mathcal{L}_{commit}$ are set to 0.5 and 0.02 respectively. For MotionLLM, we employ Gemma2-2b-it (Team et al., 2024b), a lightweight open-source LLM from Google, which offers accessibility and can be deployed on a single consumer-level GPU. The LoRA rank is set to 64 for generation and 32 for captioning, the values of alpha remain the same with the rank. All of our experiments are conducted on NVIDIA RTX4090s.

Figure 3: Motion-Agent can comprehend abstract, complex user prompts and generate accurate, long motions. It also understands and answers user questions based on real-world knowledge. Notably, the three turns in this figure stem from a continuous conversation, demonstrating the flexibility of its multi-turn capability in scenarios that should not be influenced by previous turns.

Figure 4: **Comparison with Other Methods**. Our Motion-Agent accurately generates motions involving a series of actions, while other models struggle with more complex descriptions like this, resulting in short and unclear motions.

## 4.2 RESULTS OF MOTION-AGENT

In this section, we present the results of our Motion-Agent framework, demonstrating its ability to generate long outputs through complex combinations of tasks via multi-turn conversations. It is important to note that no established ground truth exists for such tasks, aside from text-motion translation, where we do not conduct additional training for these extended tasks.

As shown in Table 1, Motion-Agent is proficient in various motion-language tasks, generating long motion sequences through natural conversational user interactions. MotionGPT (Jiang et al., 2024b) supports bidirectional translation but lacks versatility, while MoMask (Guo et al., 2024) excels in generation but is limited to this task. Although MotionChain (Jiang et al., 2024c) can perform similar functions, it requires additional datasets for task-specific instruction tuning. These methods, along with most existing approaches, are restricted to relatively short motion sequences. In contrast, without training on additional datasets, our Motion-Agent can generate longer sequences, accurately matching the given prompts, as indicated in Figure 4. While HumanML3D (Guo et al., 2022a) contains a wide range of human motions, its sequences are generally short and atomic, lasting less than 10 seconds. By decomposing descriptions of long motions into a series of short motions using LLMs and subsequently concatenating these short motions into longer sequences, our Motion-Agent can theoretically achieve infinite motion generation. This decompose-and-integrate approach can thoroughly leverage existing data for long motion generation, mapping known data distributions to unknown ones and enhancing both efficiency and scalability.

The integration of LLMs also improves the system's ability to interpret vague, abstract, or complex motion descriptions, allowing for iterative refinement through multi-turn conversations. Figure 1

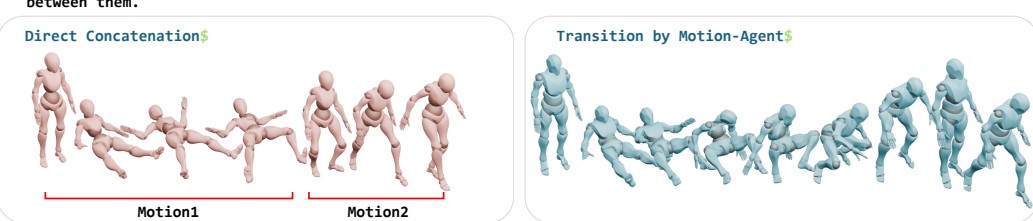

Figure 5: Motion-Agent can compose motions with smooth transitions. In this example, the two motions "a person falls down on the back" and "a person is walking" are provided to Motion-Agent in two turns. The system then generates a "stand up" motion to facilitate a seamless composition of the two motions.

already illustrates our framework's strong multi-turn contextual capabilities, enabling it to understand, extend, and edit the results of previous turns effectively. Additionally, our multi-turn functionality facilitates non-contextual requests, as evidenced by the results in Figure 3, which were generated within a single conversation comprising multiple turns. This flexibility allows users to avoid restarting for new requests. Furthermore, the results in Figure 3 demonstrate that our method can accommodate general, customized, and complex user requests through conversational and iterative exchanges. Our Motion-Agent is also capable of generating transition motions to connect and compose movements seamlessly, as shown in Figure 5, an ability that previous motion generation models struggled to achieve. This further demonstrates the motion understanding and generation capabilities of our method.

More qualitative results are presented in the appendix A.1.1 and the supplementary material.

## 4.3 EVALUATIONS OF MOTIONLLM

We evaluate MotionLLM on both text-to-motion and motion-to-text tasks to validate that it achieves satisfactory results. MotionLLM is focused on enabling bidirectional translation with minimal training load, while still maintaining competitive performance across key benchmarks. Quantitative results are shown in Table 2.

For generation, we compare our model with state-of-the-art (SOTA) approaches, including diffusion models (Tevet et al., 2023; Chen et al., 2023b; Zhang et al., 2022) and token-based models (Zhang et al., 2023b; Jiang et al., 2024b; Guo et al., 2024). Despite *fine-tuning only a small number of parameters*, our model performs competitively against these models *trained from scratch*. This demonstrates our advantages of leveraging the generalization and robustness capabilities of LLMs. Additionally, our model exhibits low MMDist, high R Precision and high Diversity, indicating strong motion-language understanding and generative capabilities. Note that MoMask (Guo et al., 2024) and the diffusion models are non-autoregressive, requiring known target lengths for generation, and evaluate using ground truth lengths. However, since the FID metric measures the distance between the distribution of generated results and ground truth, variable length generated by autoregressive models can lead to higher FID scores. Yet, our MotionLLM achieves a lower FID than some other autoregressive models such as MotionGPT (Jiang et al., 2024b) with only about one-third of trainable parameters. Additionally, the autoregressive nature of our model offers advantages over non-autoregressive models when ground truth motion lengths are not provided. This makes MotionLLM a better fit for our Motion-Agent framework, as it eliminates the need for specifying motion lengths.

In Sec. 4.4, we provide further analysis and evidence that increasing the model size can lead to overall improvements in performance scores. For a more economical choice, we selected one of the smallest LLMs (Gemma2-2B) available to the public.

For captioning, we compare models capable of bidirectional generation. Leveraging the strong text processing capabilities of LLMs, MotionLLM produces accurate descriptions of human motions. We assess the generated captions using linguistic metrics from Guo et al. (2022b), which calculate semantic similarities to ground truth captions. To ensure an accurate evaluation, we follow Jiang et al. (2024b) by using the unprocessed ground truth texts, as Guo et al. (2022b) ignores gram-

| Tasks | Methods | R Precision ↑ | | FID ↓ | MultiModal Dist ↓ | Diversity↑ |
|-------|---------|---------------|---|-------|-------------------|------------|
| | | Top 1 | Top 3 | | | |
| Generation | T2M (Guo et al., 2022a) | $0.457^{\pm.002}$ | $0.740^{\pm.003}$ | $1.067^{\pm.002}$ | $3.340^{\pm.008}$ | $9.188^{\pm.002}$ |
| | TM2T (Guo et al., 2022b) | $0.424^{\pm.003}$ | $0.729^{\pm.002}$ | $1.501^{\pm.017}$ | $3.467^{\pm.011}$ | $8.589^{\pm.076}$ |
| | *MDM* (Tevet et al., 2023) | $0.320^{\pm.005}$ | $0.611^{\pm.007}$ | $0.544^{\pm.044}$ | $5.566^{\pm.027}$ | $9.559^{\pm.086}$ |
| | *MLD* (Chen et al., 2023b) | $0.481^{\pm.003}$ | $0.772^{\pm.002}$ | $0.473^{\pm.013}$ | $3.196^{\pm.010}$ | $9.724^{\pm.082}$ |
| | *MotionDiffuse* (Zhang et al., 2022) | $0.491^{\pm.001}$ | $0.782^{\pm.001}$ | $0.630^{\pm.001}$ | $3.113^{\pm.001}$ | $9.410^{\pm.049}$ |
| | T2M-GPT (Zhang et al., 2023b) | $0.491^{\pm.003}$ | $0.775^{\pm.002}$ | $\underline{0.116}^{\pm.004}$ | $3.118^{\pm.011}$ | $\underline{9.761}^{\pm.081}$ |
| | MotionGPT (Jiang et al., 2024b) | $0.492^{\pm.003}$ | $0.778^{\pm.002}$ | $0.232^{\pm.008}$ | $3.096^{\pm.008}$ | $9.528^{\pm.071}$ |
| | MotionChain (Jiang et al., 2024c) | $0.504^{\pm.003}$ | $0.790^{\pm.003}$ | $0.248^{\pm.009}$ | $3.033^{\pm.010}$ | $9.470^{\pm.075}$ |
| | *MoMask* Guo et al. (2024) | $\mathbf{0.521}^{\pm.002}$ | $\mathbf{0.807}^{\pm.002}$ | $\mathbf{0.045}^{\pm.002}$ | $\mathbf{2.958}^{\pm.008}$ | $9.620^{\pm.064}$ |
| | **MotionLLM** | $\underline{0.515}^{\pm.004}$ | $\underline{0.801}^{\pm.004}$ | $0.230^{\pm.009}$ | $\underline{2.967}^{\pm.020}$ | $\mathbf{9.908}^{\pm.102}$ |
| | | Bleu@1↑ | Bleu@4↑ | Rouge↑ | Cider↑ | Bert Score↑ |
| Captioning | TM2T (Guo et al., 2022b) | $\underline{48.90}$ | 8.27 | $\underline{38.1}$ | 15.80 | 32.2 |
| | MotionGPT (Jiang et al., 2024b) | 48.20 | 12.47 | 37.4 | 29.20 | 32.4 |
| | MotionChain (Jiang et al., 2024c) | 48.10 | $\underline{12.56}$ | 33.9 | $\underline{33.70}$ | $\underline{36.9}$ |
| | **MotionLLM** | **54.53** | **17.65** | **48.7** | **33.74** | **42.63** |

Table 2: **Quantitative evaluation of MotionLLM on the HumanML3D (Guo et al., 2022a) test set**. For motion generation, we follow T2M (Guo et al., 2022a) for the evaluation metrics. The evaluations are conducted 20 times to obtain a 95% confidence interval. Methods indicated in *italics* utilize the ground truth lengths for estimation. Models above capable of bidirectional generation are also included in the captioning evaluation. For motion captioning, we use the ground truth captions without pre-processing and linguistic metrics suggested by Guo et al. (2022b) for evaluation. Best scores are highlighted in **boldface**, while underscore refers to the second best.

matical tense and plural forms. As demonstrated in Tab. 2, our method outperforms previous SOTA approaches across all metrics by a large margin, thanks to the language abilities of pre-trained LLMs.

## 4.4 ABLATION STUDY

**Ablation on Motion-Agent** Theoretically, the MotionLLM agent in our Motion-Agent framework can be replaced with any model capable of motion-text translation. However, models like MoMask (Guo et al., 2024), which require manual motion length input, may encounter issues (see Sec A.1.2), making autoregressive models preferable. In this study, we substitute MotionLLM with MotionGPT (Jiang et al., 2024b), which also supports bidirectional translation. After integrating with Motion-Agent, we observe that MotionGPT is capable of generating longer and more complex motions compared to its original implementation. However, it still falls short of the accuracy and smoothness achieved by using MotionLLM. For example (Figure 6), in the user prompt "A person lies face up to rest and then stands up after a while." GPT-4 decomposes this into two components: "lying face up for a while" and "transition from lying face up to standing up." While MotionGPT correctly generates the first part, it incorrectly generates the second as "from lying face down." This results in an abrupt and unsmooth transition between the two motions. In contrast, MotionLLM accurately generates both parts, ensuring a smooth, seamless motion transition.

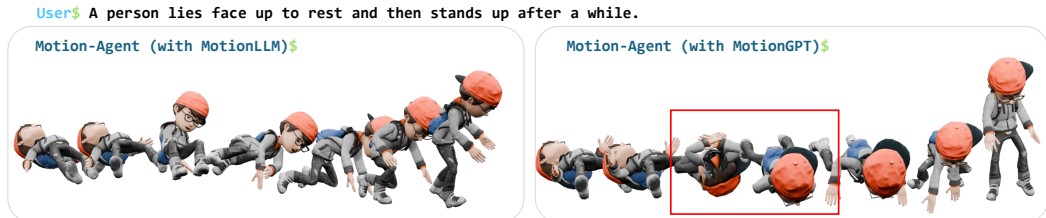

Figure 6: **Motion-Agent Ablation Study**. We substituted MotionLLM with MotionGPT and noticed that MotionGPT cannot generate smooth motion transition.

Additionally, our framework can be adapted to use different LLMs for conversation. We tested substituting GPT-4 with various models, including Llama (Touvron et al., 2023), Gemma (Team et al., 2024b), and Mixtral (Jiang et al., 2024a). Most of these models successfully generated reasonable outputs and are capable of facilitating multi-turn interactions. Some smaller models may struggle with producing the correct JSON format. This ablation study demonstrated that our framework is

applicable to all other LLMs, not just GPT-4. The performance of our framework can improve alongside the development of LLMs. More details are in Sec A.2.

Nonetheless, our framework can be summarized as a combination of a larger LLM for conversation and a motion-language translation agent, providing flexible choices for different components.

**Ablation on MotionLLM**  We conducted an ablation study to examine the impact of different LLM backbones and adapter sizes. The results are shown in Table 3, from which we may conclude that using larger backbone models or increasing the LoRA rank leads to overall improvements in the metrics. Additional ablation studies on different tokenizers and a comparison between LoRA and full fine-tuning are presented in Section A.6.

| Models | Trainable Params | R Precision ↑ | | FID ↓ | Multimodal Dist ↓ | Diversity ↑ |
| | | Top 1 | Top 3 | | | |
|---|---|---|---|---|---|---|
| T2M-GPT (Zhang et al., 2023b) | 228.4M | 0.416 | 0.745 | 0.514 | 3.007 | 10.921 |
| MotionGPT (Jiang et al., 2024b) | 220M | 0.366 | 0.680 | 0.510 | 3.527 | 10.350 |
| Gemma2-2b R=16 | 20.8M | 0.411 | 0.738 | 0.745 | 2.994 | 11.313 |
| Gemma2-2b R=32 | 41.5M | 0.415 | 0.750 | 0.712 | 2.938 | 11.251 |
| Gemma2-2b R=64 | 83.1M | 0.422 | 0.762 | 0.658 | 2.929 | 11.195 |
| LLaMA3-8B R=32 | 83.9M | 0.381 | 0.737 | 0.646 | 3.046 | 11.210 |
| Gemma2-9b R=32 | 108M | 0.439 | 0.776 | 0.438 | 2.872 | 11.151 |

Table 3: **More comparisons and ablation study on the KIT-ML (Plappert et al., 2016) dataset**. Gemma (Team et al., 2024a) and LLaMA (Touvron et al., 2023) are chosen as LLM backbones. R indicates the LoRA rank, the value of alpha is kept the same with the rank. Two other autoregressvie transformer models are included for reference.

## 5    DISCUSSION

**Limitations and Future Work.**  Our Motion-Agent specializes in generating motions of articulated 3D human body, without incorporating 3D visual understanding, such as interaction with the surrounding environment (e.g., "a person puts his hand on the table"). Also, Motion-Agent does not include detailed hand or facial movements. Nonetheless, our framework demonstrates high flexibility, making it well-suited to incorporate additional agents for handling these tasks in future extensions. Additionally, while we have conducted preliminary trials on multi-human motion generation using our Motion-Agent framework—with some initial results (see Appendix A.3)—this has not yet been fully explored. Therefore, this paper still focuses on single-human motion generation. We left the extension for human-environment interaction and multi-human interaction for future work.

**Concluding Remarks.**  In this work, we propose a novel LLM-based multimodal, conversational motion-language learning framework, offering both flexibility and generalizability. By harnessing the linguistic comprehension and generation capabilities of pre-trained LLMs, our Motion-LLM achieves strong results in bidirectional translation between motion and natural language. The Motion-Agent framework is easily expandable across various tasks through conversational interactions. Our approach is not only easy to train and adaptable but also user-friendly, making it a versatile solution for motion-language learning applications. Motion-Agent offers a comprehensive solution for enhancing LLMs' capabilities in understanding, generating, and editing human motion, aligning with our goal of teaching LLMs to interpret human motion effectively.

## ACKNOWLEDGMENTS

This work was supported in part by Dartmouth College A&S Startup fund and by the Research Grant Council of the Hong Kong SAR under Theme-based Research Scheme, grant no. T22-606/23R.

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

# A   APPENDIX

In the appendix, we present:

- Section A.1: More Qualitative Results.

- Section A.2: Ablation study on different LLMs

- Section A.3: Preliminary Trials on Multi-human Motion Generation using Motion-Agent

- Section A.4: More details about the evaluation metrics.

- Section A.5: More details regarding our implementation.

- Section A.6: More Ablation Studies on MotionLLM

## A.1   QUALITATIVE RESULTS

Rendered original videos of all examples shown in the paper can be found in the corresponding folder of the supplementary material.

### A.1.1   MOTION-AGENT

More examples of Motion-Agent are presented in Figure 7, 8, 9, and corresponding videos can be found in the supplementary material.

**User$** What is the person doing?

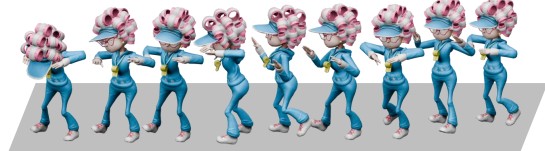

**Motion-Agent$** The person is mimicking a chicken by putting their hands to their armpits.
**User$** Do you have any ideas why he is doing like this?
**Motion-Agent$** The person could be acting like a chicken for various reasons such as performing in a play, entertaining children, participating in a game or contest, or simply having fun in a humorous or light-hearted situation.
**User$** Are there any other possibilities for this motion besides mimicking a chicken?
**Motion-Agent$** The specific description of the motion with hands placed under the armpits primarily suggests mimicking a chicken. However, it's possible, though less likely, that this motion could be used in other creative or humorous contexts not directly related to imitating a chicken, such as unconventional dance moves or as part of a physical comedy routine.
**User$** What's the difference between this motion and the previous one?

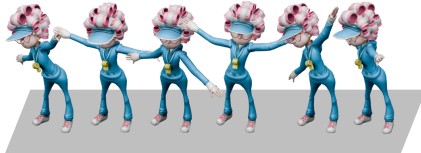

**Motion-Agent$** The difference between this motion and the previous one is that in this motion, the person explicitly raises both arms and flaps them like a bird, which suggests a broader imitation of a bird's flying motion. In contrast, the previous motion involved placing hands under the armpits and acting like a chicken, which is more specific to chicken behavior and includes mimicking the wings tucked to the sides rather than outright flapping.

Figure 7: More examples of Motion-Agent.

### A.1.2   MOTIONLLM

**Motion Generation**   Figure 10 presents the comparison on no-length-given motion generation. More qualitative results are in the supplementary material.

**Motion Captioning**   Our model is capable of generating high-quality motion captions, demonstrations are provided in Table 4.

`User$` Give me some possible scenarios for this motion.

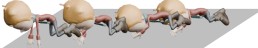

`Motion-Agent$` The person could be playing a game, sneaking up on someone, or possibly injured and trying to move to a safer location.
`User$` What do you think that person will do next?
`Motion-Agent$` After crawling forward, the person might try to stand up, continue crawling, or reach for something in front of them. The exact next action would depend on the context of why they are crawling.
`User$` Imagine this person is in one of the scenarios you just said, and predict a longer motion.
`Motion-Agent$`

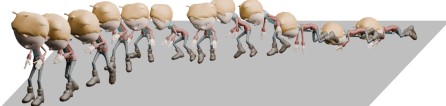

`User$` Can you explain it?
`Motion-Agent$` In the scenario where the person is sneaking up on someone or trying to move cautiously, the sequence of motions could be as follows: First, the person is slowly crawling forward. Then, they stand up from the crawling position. After standing up, they look around cautiously to ensure the environment is safe or to locate their target. Finally, they start walking slowly to continue their cautious movement.
`User$` Then let's say this person is noticed and ran away. Generate a longer motion.
`Motion-Agent$`

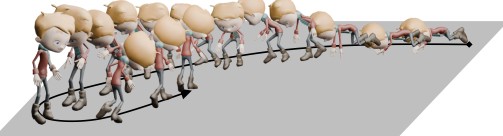

`User$` Can you briefly write in the style of a news report about what's happening?
`Motion-Agent$` In a surprising turn of events, an individual was spotted engaging in what appeared to be a stealthy approach, slowly crawling forward in an attempt to sneak up on another person. Witnesses reported that the sneaking individual stood up cautiously after a brief moment, looking around to assess the situation. However, the stealthy maneuver did not go unnoticed. Upon realizing they had been detected, the individual quickly turned and fled the scene, leaving onlookers bewildered by the unexpected encounter.

Figure 8: More examples of Motion-Agent.

| Motion | Model | Caption |
|---|---|---|
| | Ground Truth | a person walks forward just like a mummy |
| | TM2T | a person walk in a counterclockwise circle with their arm out to the side |
| demo_1 | MotionGPT | the person is walking like a mummy from the dead |
| | Ours | a person walks forward while holding arms out as if to be a zombie |
| | Ground Truth | a person walks forward slowly while their right hand is slightly elevate |
| | TM2T | a person slowly walk forward while hold onto something with their left hand |
| demo_2 | MotionGPT | a person walks forward slowly, placing one foot in front of the other, on a belt that circulates, enabling the person to effectively slowly walk in place. |
| | Ours | the person is walking on a balance beam |
| | Ground Truth | a person moves side to side in a zigzag fashion backwards |
| | TM2T | a person does a cartwheel to the right |
| demo_3 | MotionGPT | a person is practing defense moves. |
| | Ours | a person walks backwards in zig-zag motion |
| | Ground Truth | a person makes and drinks a cup of coffee |
| | TM2T | person hold something with their right hand and make a sawing motion with their left hand |
| demo_4 | MotionGPT | a person is eating something |
| | Ours | a person uses their left hand to open a bottle, drinks from it, then places the bottle back down |

Table 4: Comparsion of motion captioning ability across different models. Original motions can be found in supplementary material.

## A.2 ABLATION STUDY ON DIFFERENT LLMs

In this study, we replace GPT-4 with several other LLMs, including Llama (Touvron et al., 2023), Gemma (Team et al., 2024b), and Mixtral (Jiang et al., 2024a). The experiment involved a straight-forward two-turn conversation. In the first turn, the we prompted, "Generate a motion that a person

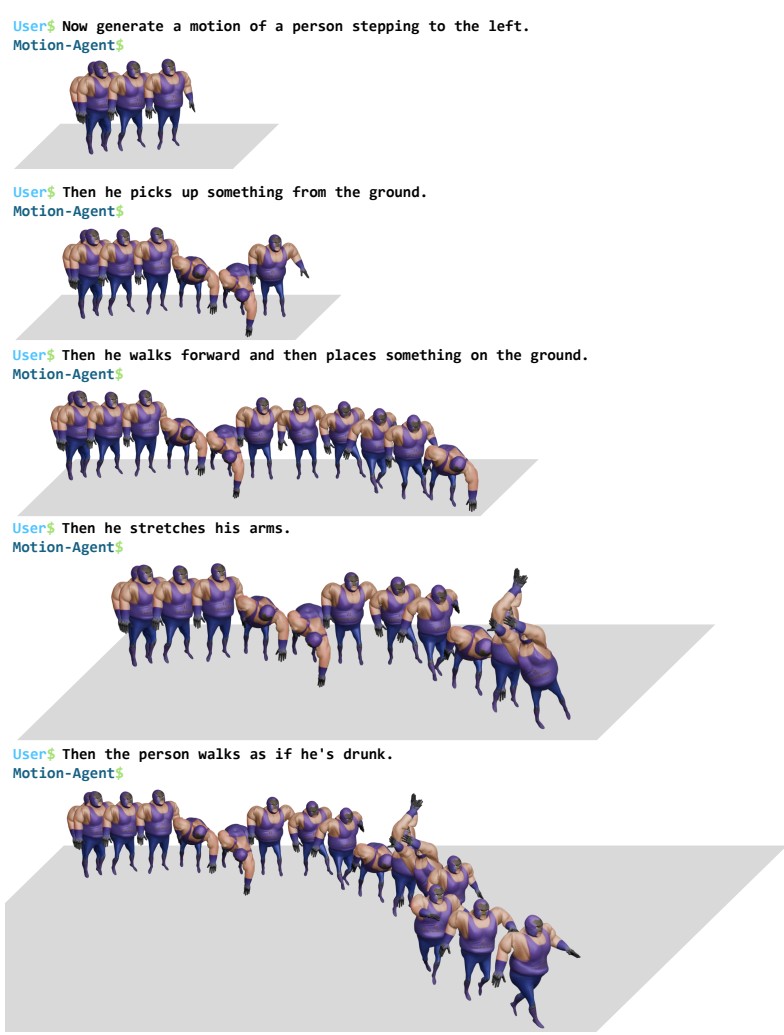

Figure 9: More examples of Motion-Agent.

is doing exercise." In the second turn, we provided a motion that a person is slowly crawling forward and asked, "Briefly explain the possible scenarios for this motion." The decomposed arguments from the agent in the first turn and the response from the second turn are presented in Table 5.

Overall, we observe that different LLMs can generate reasonable outputs in response to user requests. However, smaller models, such as Llama-3-7B and Mixtral-8x7B, while capable of producing some acceptable responses, struggled to adhere strictly to the instructed JSON format. As a result, the agent was unable to parse their outputs successfully.

### A.3 MULTI-HUMAN WITH MOTION-AGENT

In this section, we present the results of our preliminary trials on multi-human motion generation using the Motion-Agent framework, specifically focusing on generating motions for two individuals.

In our implementation, each person is represented in the HumanML format (Guo et al., 2022a), with their motions defined separately. To uniquely define the motions of both individuals, we incorporate location information for the first frame, represented by a tuple of three parameters, relative $r = (\theta, x, z)$. Here, the first person is always positioned at the origin in 3D space, and the relative tuple $r$ determines the position of the second person concerning the first. The parameter $\theta$ denotes the rotation radius, while $x$ and $z$ represent the coordinates (with the y-axis as vertical). Therefore, the

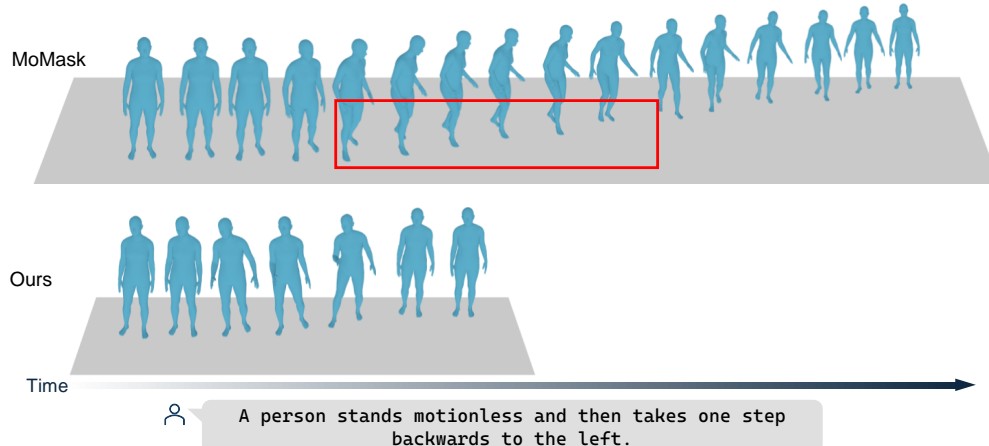

MoMask

Ours

Time

A person stands motionless and then takes one step backwards to the left.

Figure 10: Comparison between MotionLLM and MoMask (Guo et al., 2024), which is non-autoregressive. During regular inference, MoMask uses a length estimator to predict the length conditioned on the text. This estimator is likely to fail. In this example, their incorrect predicted length causes severe drifting.

| LLM | Turn 1 Decomposition Results | Turn 2 Reasoning Response |
|---|---|---|
| GPT-4-Turbo | 'A person does jumping jacks.', 'A person does push-ups.', 'A person does sit-ups.', 'A person stretches.' | "The person could be participating in a military training exercise, playing a game that involves crawling, navigating through a confined space, or engaging in a fitness routine that includes floor-based movements." |
| Llama-3-70B | 'A person stands up straight.', 'A person raises their arms above their head.', 'A person bends their knees slightly.', 'A person starts jogging in place.', 'A person lowers their arms to their sides.' | "The person could be participating in a military or law enforcement training exercise, engaging in a crawling competition, playing a game that involves crawling, or navigating through a confined or obstacle-filled space." |
| Gemma2-27B | 'A person does jumping jacks.', 'A person does push-ups.', 'A person does sit-ups.' | Possible scenarios include: army training exercise, searching for something lost on the ground, playing a game, navigating a tight space, or enacting a dramatic scene. |
| Llama-3.1-8B | 'A person starts in a standing position.', 'A person bends their knees.', 'A person lowers their body.', 'A person stands up.', 'A person repeats the motion.' | "The person could be crawling through a tight space, crawling on the floor for exercise, or crawling as a form of slow and deliberate movement in a situation where walking or running is not feasible." |
| Llama-3-7B | Failed to follow the format | Failed to follow the format |
| Mixtral-8x7B | Failed to follow the format | Failed to follow the format |

Table 5: Comparison of Using Different LLMs on a Two-Turn Prompt.

Figure 11: Multi-human Motion Generation using Motion-Agent.

motion of each person together with $r$ can uniquely determine the whole motion. In this context, GPT-4 is tasked with generating three outputs: the arguments for MotionLLM for each person and the relative tuple $r$.

Figure 11 shows an example of multi-human generation using Motion-Agent. In this case, the two arguments are "A person waves.", and $r = (3.14, 0, 1)$, indicating that the second person rotates 180 degrees (since $3.14 \approx \pi$) from facing the $z^+$ direction (hence positioned face to face with the first person) and is standing 1 meter away from the first person.

## A.4 EVALUATION METRIC

We detail the calculation of several evaluation metrics proposed in Guo et al. (2022a). We denote ground-truth motion features, generated motion features, and text features as $f_{\text{gt}}$, $f_{\text{pred}}$, and $f_{\text{text}}$. Note that these features are extracted with pretrained networks in Guo et al. (2022a).

**Multimodal Distance (MM-Dist).** MM-Dist is widely used to evaluate the motion generation ability of the model. MM-Dist measures the distance between the text embedding and the generated motion feature. Given $N$ randomly generated samples, the MM-Dist measures the feature-level distance between the motion and the text. It computes the average Euclidean distances between each text feature and the generated motion feature from this text:

$$\text{MM-Dist} = \frac{1}{N} \sum_{i=1}^{N} \| f_{\text{pred},i} - f_{\text{text},i} \|$$

where $f_{\text{pred},i}$ and $f_{\text{text},i}$ are the features of the $i$-th text-motion pair.

**Frechet Inception Distance (FID).** FID measures the distance of motion features distribution between real and generated motions. We calculate FID by

$$FID = \| \mu_{\text{gt}} - \mu_{\text{pred}} \|^2 - \text{Tr}(\Sigma_{\text{gt}} + \Sigma_{\text{pred}} - 2(\Sigma_{\text{gt}} \Sigma_{\text{pred}})^{1/2})$$

where $\mu_{\text{gt}}$ and $\mu_{\text{pred}}$ are the means of $f_{\text{gt}}$ and $f_{\text{pred}}$. $\Sigma$ is the covariance matrix and $\text{Tr}$ denotes the trace of a matrix.

**R precision** Given the motion sequence and 32 text descriptions (1 ground-truth and 31 randomly selected mismatched descriptions), we rank the Euclidean distances between the motion and text embeddings to get Top-1, Top-2, and Top-3 accuracy of motion-text;

**Diversity.** Diversity measures the variance of the whole motion sequences across the dataset. We randomly sample $S_{\text{dis}}$ pairs of motion and each pair of motion features is denoted by $f_{\text{pred},i}$ and $f'_{\text{pred},i}$. The diversity can be calculated by

$$Diversity = \frac{1}{S_{\text{dis}}} \sum_{i=1}^{S_{\text{dis}}} \| f_{\text{pred},i} - f'_{\text{pred},i} \|$$

In our experiments, we set $S_{\text{dis}}$ to 300 as (Guo et al., 2022a).

**Linguistic metrics.** Linguistic metrics including Bleu (Papineni et al., 2002), Rouge (Lin, 2004), Cider (Vedantam et al., 2015) and Bert Score (Zhang et al., 2020), we follow TM2T (Guo et al., 2022b), using NLPEval to calculate. Readers can refer to their papers for further details.

## A.5    MORE IMPLEMENTATION DETAILS

**Prompts For MotionLLM.**    We use different prompts for different tasks.

| Task | Prompts |
|---|---|
| Motion Generation | Generate a motion matching the following input human motion description. |
| Motion Captioning | Generate a caption matching the following input human motion token sequence. |

Table 6: Instructing prompts for MotionLLM training and inference.

**Hyper-parameters.**    Our hyper-parameters settings for different tasks.

| Hyper-parameter | Motion Generation | Motion Captioning |
|---|---|---|
| Batch size | 6 | 6 |
| Learning rate | 1e-5 | 1e-5 |
| LoRA rank | 64 | 32 |
| LoRA alpha | 32 | 32 |
| LoRA dropout | 0.1 | 0.1 |
| Codebook size | 512 | 512 |
| Codebook dim | 512 | 512 |
| Total vocab size | 256514 | 256514 |

Table 7: Hyper-parameters of our models used in our main experiments. Other VQ training settings are borrowed from T2M-GPT (Zhang et al., 2023b)

## A.6    MORE ABLATION STUDIES ON MOTIONLLM

**Ablation study on different tokenizers**    We conducted additional experiments replacing our current VQ-VAE tokenizer with the RVQ-VAE tokenizer used in MoMask (Guo et al., 2024). RVQ-VAE differs from traditional VQ-VAE by using $Q+1$ ordered codebooks. The first codebook is used for base tokens, similar to traditional VQ, while the remaining $Q$ codebooks are used to represent residuals for enhancing fidelity.

Following MoMask's approach, the MotionLLM in our case is aware of the base layer tokens. To predict the residual layer token sequences, we follow MoMask to use an additional non-autoregressive (NAR) transformer that takes as input the base layer token sequences from the LLM output. This NAR transformer can thus be considered as part of our detokenizer, which does not affect the LLM's inference or training process.

The results are shown in Table 8. Our results show that using RVQ-VAE does not significantly improve performance compared to the original VQ-VAE, validating that the VQ-VAE model is already sufficiently effective for our tasks. While RVQ-VAE provides some potential improvements in fidelity, it introduces additional training overhead (the 13.4M parameters are from the NAR transformer). Specifically, the NAR transformer required for residual token prediction is challenging to train and adds complexity to the system (namely, requiring an additional training step for the NAR transformer).

Notwithstanding, we believe that both VQ-VAE and RVQ-VAE are effective for motion tokenization, and that the choice between them is more of an engineering decision based on the specific trade-offs involved. Our Motion-Agent framework remains flexible and can support different tokenizers, as long as they can convert motions into discrete representations. Future advancements in tokenization methods may further enhance the framework's performance and capabilities.

| Models | Trainable Params | R Precision ↑ | | FID ↓ | Multimodal Dist ↓ | Diversity ↑ |
|---|---|---|---|---|---|---|
| | | Top 1 | Top 3 | | | |
| Gemma2-2b R=32 VQ | 41.5M | 0.415 | 0.750 | 0.712 | 2.938 | 11.251 |
| Gemma2-2b R=64 VQ | 83.1M | 0.422 | 0.762 | 0.658 | 2.929 | 11.195 |
| Gemma2-9b R=32 VQ | 108M | 0.439 | 0.776 | 0.438 | 2.872 | 11.151 |
| Gemma2-2b R=64 RVQ | 83.1M+13.4M | 0.429 | 0.768 | 0.647 | 2.857 | 10.126 |
| Gemma2-2b Full VQ | 2697.4M | 0.423 | 0.774 | 0.591 | 2.913 | 11.138 |

Table 8: More ablation studies conducted on KIT-ML (Plappert et al., 2016).

**Ablation study on LoRA vs Full fine-tuning** We also experimented fine-tuning all the parameters of LLM, the corresponding result is shown in Table 8. When using a 2B backbone, full fine-tuning does improve overall scores but at the costly expense of significantly increasing training overhead, which is in stark contrast to LoRA fine-tuning a 9b backbone model which requires over 20 times fewer parameters trained. Fully training such a large model also demands substantially more computational power and memory, making it nearly impossible on consumer-level GPUs. This result indicates that LoRA is indeed a more efficient and effective approach than full fine-tuning, suggesting it is a better alternative for scaling up to a larger backbone in the future.

