# OpenReview forum: "Motion-Agent: A Conversational Framework for Human Motion Generation with LLMs"
_ICLR.cc/2025/Conference — ICLR 2025 Poster_

### Official Review · Reviewer_oMaw · 2024-10-28

**Soundness:** 3
**Presentation:** 3
**Contribution:** 3
**Rating:** 6
**Confidence:** 2

**Summary:**

This paper introduces Motion-Agent, an efficient framework for general human motion generation, and it also develops a generative agent, Motion LLM, which bridges the gap between motion and text. Experiments show that with few-shot fine-tuning, MotionLLM can achieve state-of-the-art performance. Motion-Agent can generate complex motion sequences.

**Strengths:**

1. This paper presents a framework to bridge text and motion. This part is novel in the opinion of the reviewer. The proposed motion tokenization approach and Motion-language Agent have some level of novelty.

2. Table 1 summarizes the advantages of this work against several recent competitors. It seems that this work supports multi-turn editing, reasoning, and composition. Also, Motion LLM can generate longer motion, which is superior to previous works.

3. Motions in this work and its supplementary (including video results) look cool. This indicates that the proposed model can generate complex motions.

**Weaknesses:**

1. The website in the supplementary cannot function well (videos cannot be rendered). The reviewer suggests authors publish their websites on anonymous hosts (such as Google Pages). Moreover, it's better to show case more diverse prompts.

2. More ablation studies are welcomed. For example, different types of tokenizers, detailed comparison between LLaMa and Gemma, etc.

**Questions:**

1. What's the rendering engine to generate those visualizations?

2. Can this framework benefit down stream tasks such as robotics and policy learning?

---

> ### Author Response · Authors · 2024-11-22
> **Reply to Reviewer oMaw**
>
> We thank you for your thoughtful reviews and suggestions.
>
> We enthusiastically encourage the reviewer to watch our demos which present in our opinion some of the most impressive results to date. We tested on different machines and Windows PCs should be most stable and have no problem loading the videos. Please be reminded that the zip file has to be expanded and unzipped before clicking "index.html"; Common browsers such as the latest Chrome should work. On the other hand, the videos can be directly available in the folders.
>
> Thank you for your suggestions and we are committed to making a more stable page with more diverse demos.
>
> More ablation studies are provided, please refer to the general response.
>
> **Render Engine**
>
> After Motion-Agent generates the joint representations, they are converted into motion capture files. We then map the generated motion data to a randomly selected 3D avatar in *BLENDER* to form an animated character. Finally, we render the video in *BLENDER* for visual demonstration.
>
> **Can this framework benefit downstream tasks such as robotics and policy learning?**
>
> We believe our framework has the potential to contribute to the fields of robotics and policy learning, which we also plan to explore in future work. The current motion representation is based on human joint-level data, which may be adapted for humanoid robot control. Leveraging LLM offers an attractive alternative to robotic control, navigation, and manipulation.

---

> > ### Comment · Reviewer_oMaw · 2024-11-22
> > **Thank you for the rebuttal.**
> >
> > Final decisions will be made after the discussion with other reviewers.

---

### Official Review · Reviewer_QaCH · 2024-11-03

**Soundness:** 3
**Presentation:** 3
**Contribution:** 3
**Rating:** 6
**Confidence:** 5

**Summary:**

Motion Agent introduces a conversational framework for generating, editing, and reasoning about human motion in multiple steps. This framework is built on a MotionLLM fine-tuned with LoRA. With a motion tokenizer-detokenizer and a text detokenizer, Motion Agent can create new motions and generate text that explains or describes the motions in natural language. Fine-tuning the large LLM allows Motion Agent to generalize better than current state-of-the-art methods, produce longer motion sequences, and handle language-guided edits.

**Strengths:**

1) The paper is well-written and supported by experiments, with a good presentation of ablation studies and applications. The supplementary material includes videos for temporal generations, which helps in reviewing the work better.

2) This research is timely, as using LoRA fine-tuning on LLMs for multimodal human motion generation allows for models that don’t rely on task-specific data.

3) Multi-turn editing and the ability to generate long motion sequences are valuable directions for the field.

**Weaknesses:**

1) Lines 280-282: Multiple adapters are mentioned to work together, and fig 2 shows two LoRA adapters after fine-tuning. It looks like they’re applied to the base LLM in separate branches. How does MotionLLM.generate or MotionLLM.caption work—are they called every time during inference, or just when needed? This information would help clarify the pipeline.

2) Line 323 states that there is no ground truth for these tasks, but the videos show apparent issues like body interpenetration, lack of physical constraints (e.g., floor contact), and body deformation. There are also no quantitative metrics for multi-turn edited generations. Even though the motion quality seems better than the current SOTA, these limitations stand out. One way to address this would be to explain why these interpenetrations occur, especially since the motion tokenizer/detokenizer was trained on motion reconstruction losses, which should ideally prevent this. Another approach could be to generate multi-turn prompts based on test set motion sequences and provide metrics like FID, where the test sample could act as a pseudo ground truth.

3) The examples in fig 3 show different genders/body types across three instances. How does Motion Agent choose each identity or gender? Are all identities compatible in terms of body kinematics, or could retargeting cause issues if aiming for consistent identity? This needs more explanation.

4) Figure 5 shows in-betweening, but there’s no quantitative metric to evaluate it. Using a metric like NPSS (e.g., https://arxiv.org/pdf/1809.03036, https://arxiv.org/pdf/2102.04942) could provide useful information on the in-between motion quality.

5) Tab 2 - Generation Task: FID scores are lower than the state-of-the-art. As noted in Lines 416-418, could the motion length be constrained to match other methods to provide a clearer comparison of the FID scores?

**Questions:**

1) What are the benefits of using VQVAE over other models like diffusion/VAE?

2) Which body joint representations were used for training?

3) The extended motion sequences keep the original motion unchanged. This is visible in all the provided videos. For example, with prompt p1, motion m1 is generated; when p2 is added, it generates a new motion m2, so the combined motion is m1 + m2. How is m1 preserved—does the new generation only respond to the new prompt?

4) Lines 191-192 mention that the motion detokenizer is important for smoothing transitions between different motion sequences. Are the motions generated by MotionLLM concatenated and sent to the motion detokenizer after each prompt?

---

> ### Author Response · Authors · 2024-11-22
> **Reply to Reviewer QaCH (part1/2)**
>
> We thank you for your thoughtful reviews and suggestions.
>
> **How does MotionLLM.generate or MotionLLM.caption work—are they called every time during inference, or just when needed?**
>
> GPT-4 will decide whether to call MotionLLM in each turn. So they will only be called when needed.
>
> **Interpenetration**
>
> We would like to clarify that the apparent issues of body interpenetration and lack of physical constraints arise due to the graphical rendering process, which is a separate step after motion generation. Specifically, after Motion-Agent generates the joint representations, they are converted into motion capture files. We then map the generated motion data to a randomly selected 3D avatar in *BLENDER* (graphics package) for visualization purposes.
>
> However, 3D avatars come in various shapes and sizes, and while the generated joint movements follow physical constraints, the mapping of these joints to the avatar’s mesh may cause interpenetration, especially when the avatars have differing proportions. We acknowledge that this graphical issue is not ideal, but it stems from the complexities of avatar's discrete mesh geometry for
>  skinned animation, rather than the motion generation process itself.
>
> These graphical or visualization inconsistencies may be resolved by relevant future graphics research, by incorporating better avatar normalization and collision handling techniques. We believe this will reduce the likelihood of interpenetration and improve the overall visual quality of the generated motions.
>
> **Different genders/body types**
>
> They are arbitrarily chosen just for the purpose of visualization. The raw output consists of just the coordinates of the pertinent joints.
>
> **Motion In-between**
>
> We believe Figure 5 does not accurately represent the other "in-between" task that predicts motions (that best interpolate) between keyframes. Instead, we illustrate motion composition, where direct concatenation may result in unnatural transitions. Motion-Agent addresses this problem by generating a semantically coherent motion (specifically, by using GPT-4 to create intermediate motion descriptions) to smoothly connect adjacent segments.
>
> Notwithstanding, we heeded the suggestion and finetuned MotionLLM to perform the "in-between" task, and the results are provided below. However, we would like to emphasize that this is not the primary focus of our paper. Additionally, we believe it is unnecessary to empower MotionLLM with these capabilities, as GPT-4 is more effective at generating new motion captions for both prediction and in-between motion generation.
>
> ||Motion Prediction||||Motion In-between|||
> |-|-|-|-|-|-|-|-|
> |**Methods**| FID&nbsp;↓        | Diversity&nbsp;↑ | ADE&nbsp;↓  | FDE&nbsp;↓  | FID&nbsp;↓        | Diversity&nbsp;↑ | ADE&nbsp;↓  |
> | MDM       | 6.031             | 7.813          | 5.446       | 8.561       | 2.698             | 8.420          | 3.787       |
> | MotionGPT | 0.905             | 8.972          | 4.745       | 6.040       | **0.214**         | 9.560          | **3.762**   |
> | MotionLLM | **0.716**         | **9.836**      | **4.641**   | **5.057**   | 0.286             | **9.678**      | 3.789       |
>
> *Comparison of motion prediction and motion in-between on part of AMASS dataset.*
>
> **Constrain length for comparison**
>
> We appreciate the reviewer’s suggestion to constrain the motion length for a more consistent comparison with other methods. However, we respectfully disagree with imposing a fixed motion length as a constraint for evaluating generated motion. The length of motion inherently varies based on the type of action being generated—some motions are brief, lasting less than a second, while others require longer trajectories for accurate representation.
>
> Imposing a fixed temporal window, similar to those used in motion recognition tasks, would not be ideal. Although some datasets use fixed-length videos for motion recognition, they often introduce problems such as irrelevant short motions or forced cropping of longer motions, which can compromise the evaluation process. Thus, we believe that allowing variable motion lengths provides a more realistic and meaningful comparison across different methods. Moreover, the ability to generate variable-length motions should be viewed as an advantage rather than a disadvantage, as it demonstrates greater flexibility and adaptability in motion generation.

---

> ### Author Response · Authors · 2024-11-22
> **Reply to Reviewer QaCH (part2/2)**
>
> **What are the benefits of using VQVAE over other models like diffusion/VAE?**
>
> We believe this question is similar to a question asking why diffusion/VAE-based models are not used in the next-token prediction in language models (LLMs). While diffusion and VAE models have shown impressive results in text-to-motion generation, we argue that if a robust and accurate next-token prediction method exists for motion generation, these models may not be necessary. While diffusion and VAE models can indeed be used as post-processing tools to further refine the generated motion, our experiments suggest that they are not essential for achieving high-quality motion generation in our framework.
>
> On the other hand, a key benefit of using VQVAE over models such as diffusion or VAE is its support for bidirectional translation, which allows for both generation and captioning tasks. In contrast, diffusion models are primarily designed for generation tasks and do not natively support the same level of flexibility for captioning or other reverse tasks. This bidirectional capability of VQVAE-based methods makes it particularly suitable for tasks that require both understanding and generating motion, such as the motion-language framework we propose, where both motion generation and motion-to-text conversion are critical.
>
> **Which body joint representations were used for training?**
>
> We use the HumanML representation.
>
> Reference: Guo, Chuan, et al. "Generating diverse and natural 3d human motions from text." Proceedings of the IEEE/CVF Conference on Computer Vision and Pattern Recognition. 2022.
>
> **How is m1 preserved—does the new generation only respond to the new prompt?**
>
> In our approach, both motions $m_1$ and $m_2$ are represented as token sequences before they are decoded into motion representations. Specifically, let us assume that motion $m_1$ corresponds to token sequence $s_1$, and motion $m_2$ corresponds to $s_2$, such that $m_1 = D(s_1), m_2 = D(s_2)$, where $D$ is the detokenizer.
>
> In the first turn, when the prompt $p_1$ is provided, the token sequence $s_1$ is generated. This sequence can then be preserved and temporarily stored. In the second turn, when prompt $p_2$ is added, the new token sequence $s_2$ is generated. To combine the two motions, we concatenate the sequences into $s = (s_1, s_2)$, which is then decoded into the final motion sequence.
>
> While the motions themselves are not directly preserved, the token sequences are preserved. Since $s_1$ is preserved and used in the concatenation, the corresponding motion $m_1$ will remain almost identical to its original form. Thus, while the motion sequence is extended with new content, the original motion is effectively preserved in the final generated sequence.
>
> On the other hand, if such preservation is not desired, a new $s_1$ can be generated in the second turn, which might result in a different motion and cost additional computation.
>
> **Are the motions generated by MotionLLM concatenated and sent to the motion detokenizer after each prompt?**
>
> If the user asks for generation, GPT-4 may call MotionLLM multiple times, and MotionLLM will generate corresponding motion tokens. After generating all tokens, they are concatenated and decoded together by the decoder.

---

> > ### Comment · Reviewer_QaCH · 2024-11-28
> > **Response to rebuttal**
> >
> > Thank you for your comprehensive answers, it has adequately addressed my concerns.

---

### Official Review · Reviewer_DEqA · 2024-11-04

**Soundness:** 3
**Presentation:** 3
**Contribution:** 3
**Rating:** 5
**Confidence:** 3

**Summary:**

The paper introduces Motion-Agent, a novel conversational framework for 3D human motion generation, editing, and understanding that leverages large language models (LLMs). The framework employs a pre-trained language model called MotionLLM, which bridges the gap between motion and text by encoding and quantizing motions into discrete tokens aligned with the language model's vocabulary. By fine-tuning only 1–3% of the model's parameters using adapters, MotionLLM achieves performance comparable to diffusion models and other transformer-based methods trained from scratch. Integrating MotionLLM with GPT-4 without additional training, Motion-Agent enables the generation of highly complex motion sequences through multi-turn conversations, a capability previous models lacked. The framework supports a wide range of motion-language tasks, offering versatile capabilities for generating and customizing human motion through interactive conversational exchanges. Experiments demonstrate that Motion-Agent effectively handles intricate, multi-turn interactions and achieves strong results in both motion generation and captioning tasks.

**Strengths:**

Originality: The paper presents an innovative approach by integrating large language models into 3D human motion generation, a relatively unexplored area. The conversational framework allowing multi-turn interactions for motion generation and editing is particularly novel.
Quality: The proposed method effectively leverages pre-trained LLMs with minimal fine-tuning, demonstrating competitive performance with significantly reduced computational resources compared to models trained from scratch.
Clarity: The methodology is well-articulated, with clear explanations of the motion tokenizer/detokenizer and how MotionLLM integrates with GPT-4. The inclusion of qualitative examples and ablation studies aids in understanding the framework's capabilities.
Significance: The ability to generate complex, customizable human motions through conversational interactions has substantial implications for animation, virtual reality, and human-computer interaction. The framework's versatility enhances its practical relevance in various applications.

**Weaknesses:**

Evaluation Metrics: The paper primarily presents qualitative results and some standard metrics but lacks comprehensive quantitative comparisons with state-of-the-art models on standardized benchmarks, making it challenging to fully assess performance improvements.
Limited Ablation Studies: While an ablation study is included, more extensive experiments isolating the contributions of each component (e.g., the impact of different LLMs, the motion tokenizer) would strengthen the evaluation.
Dependence on Proprietary Models: Utilizing GPT-4 as the conversational LLM may limit reproducibility and accessibility, as GPT-4 is not openly available to all researchers.
Scalability and Efficiency: The paper does not thoroughly discuss the computational efficiency and real-time performance of the framework, which is critical for interactive applications.

**Questions:**

Can the authors provide more quantitative evaluations comparing Motion-Agent with state-of-the-art models on standard benchmarks?
How does the choice of the pre-trained LLM (e.g., GPT-4 vs. open-source models) impact the performance of the framework? Have experiments been conducted with other LLMs?
What are the computational requirements (e.g., inference time, memory usage) of Motion-Agent, especially for real-time interactive applications?
How does the framework handle motions involving interactions with the environment or multiple agents (e.g., multi-human interactions)?
Are there limitations regarding the diversity or realism of the generated motions when handling highly complex or extended sequences?

**Details Of Ethics Concerns:**

no needed

---

> ### Author Response · Authors · 2024-11-22
> **Reply to Reviewer DEqA (part1/2)**
>
> We thank you for your thoughtful reviews and suggestions.
>
> For the ablation study on different tokenizers, please refer to the general response.
>
> **Standard benchmarks**
>
> The success of Motion-Agent relies on the robust bi-directional translation capabilities of MotionLLM and the well-known conversational and generalization strengths of LLMs.
> As presented in the paper, we have compared MotionLLM with SOTA models, demonstrating that MotionLLM performs competitively in bi-directional generation tasks. For conversational tasks involving Motion-Agent, the generated motions are derived from concatenating MotionLLM's outputs without additional training. To our knowledge, no standard benchmarks currently exist for such complex, flexible, multi-turn conversational motion generation tasks.
>
> **Different LLMs**
>
> We have conducted experiments with different LLMs in simple two-turn conversational prompts, detailed in Appendix A.2. In these experiments, we substituted GPT-4 with open-source models (Llama, Gemma, and Mixtral) to examine any impact on performance. Our observations indicate that the general performance remains consistent, suggesting that Motion-Agent’s core functionality is robust across different LLM choices. We attribute this consistency to the simplicity of the required understanding and planning tasks, which are not overly demanding for modern LLMs. Additionally, models having been trained for JSON output may better generate the correct format. We welcome suggestions for additional evaluation approaches to further explore the influence of different conversational LLMs on Motion-Agent.
>
> GPT-4 and other conversational LLMs are used without modification, and we plan to open-source our prompts and implementation in the future. Both the API and the web-based versions of these models are sufficient for reproducing our results.
>
> The following provides more comparisons between GPT-4 and Llama 3.1 8B. Our findings indicate that a lightweight, open-source LLM can achieve results comparable to GPT-4. GPT-4 may tend to produce more accurate and reasonable motion descriptions and slightly better generation results with MotionLLM, but the observed differences are relatively minor:
>
> *A football player is doing some warm-ups on the pitch.*
>
> GPT-4:
>   1. MotionLLM.generate(`A person stretches their legs.');
>   2. MotionLLM.generate(`A person performs jumping jacks.');
>   3. MotionLLM.generate(`A person jogs in place.');
>   4. MotionLLM.generate(`A person kicks the ball lightly as a warm-up.')
>
> Llama 3.1 (8B):
>   1. MotionLLM.generate(`A person runs in place.');
>   2. MotionLLM.generate(`A person jumps up and lands.');
>   3. MotionLLM.generate(`A person stretches their arms overhead.');
>   4. MotionLLM.generate(`A person bends to touch their toes.')
>
>
> *A person seems to be shaking something like a spray can, then starts punching something, and raises both arms above the head while stepping to the left.*
>
> GPT-4:
>   1. MotionLLM.generate(`A person shakes a can.');
>   2. MotionLLM.generate(`A person punches forward repeatedly.');
>   3. MotionLLM.generate(`A person raises both arms above their head.');
>   4. MotionLLM.generate(`A person steps to the left while raising both arms above their head.')
>
> Llama 3.1 (8B):
>   1. MotionLLM.generate(`A person shakes something like a spray can.');
>   2. MotionLLM.generate(`A person punches something.');
>   3. MotionLLM.generate(`A person raises both arms above the head.');
>   4. MotionLLM.generate(`A person steps to the left.')
>
> **Computational requirements**
>
> MotionLLM uses Gemma2-2B as its backbone, one of the smallest publicly available LLMs, which enables it to run efficiently on a single consumer-level GPU without requiring additional optimizations such as quantization.
>
> For the conversational LLMs, if APIs are used, the computational overhead is handled remotely. We have also experimented with other models such as Llama-8B and Gemma2-27B, both of which can be deployed locally on servers with reasonable computational resources.
>
> The inference time of Motion-Agent aligns with that of the underlying LLMs, and can benefit from typical LLMs optimizations. Given the generally acceptable speed of GPT-4 and other LLMs, our system's inference time is consistent with user expectations for real-time interaction.

---

> ### Author Response · Authors · 2024-11-22
> **Reply to Reviewer DEqA (part2/2)**
>
> **Multi-human and human-environment interaction**
>
> Though we leave this as future work which is not yet adequately explored, we can provide some insight into how to extend Motion-Agent to handle such scenarios. The core idea is to introduce additional agents (to work with the current MotionLLM agent), to assist these tasks.
>
> For multi-human motion generation, we have presented a simple example in Appendix A.3. The idea is to decompose the interaction into two individual motions, each generated separately. More specifically, another coordinating agent can be introduced to synchronize these individual motions or enforce more physical constraints.
>
> For human-environment interactions, the approach is similar. Additional agents can be introduced to enable the system to understand the scene context for track objects, ensuring that the motions align with environmental changes or object movements.
>
> Though not fully explored, this agent-based approach allows for scalability and adaptability to more complex scenarios involving multiple humans or environmental factors.
>
> **Are there limitations regarding the diversity or realism of the generated motions when handling highly complex or extended sequences?**
>
> When given highly complex or long prompts, such as movie scripts describing complex action sequences and responses, Motion-Agent may not always produce satisfactory results to the users in a single turn. However, this is exactly where multi-turn capabilities may help. As the generation process can be iterative, one can refine and edit the output through multiple turns, enabling the generated motion to satisfy the user’s request for fast and easy convergence to final outcome, as demonstrated in Figure 1.

---

### Official Review · Reviewer_7jDa · 2024-11-04

**Soundness:** 4
**Presentation:** 4
**Contribution:** 3
**Rating:** 8
**Confidence:** 4

**Summary:**

This work presents a framework capable of understanding, generating, and editing 3D human motion based on dialogue. Specifically, the model includes a more advanced LLM (GPT-4) play a role of "coordinator" and a motion/language translator fine-tuned from a lightweight language model (LoRA). The experiments (including the ablation study) are thorough, and the results (including the demos) demonstrate excellent performance.

From my perspective, the reason for achieving this performance isn’t due to improvements in the language-motion model itself, but is because the coordinator achieves the high level of understanding, decomposing, and recording the tasks before language-motion model, significantly enhancing the results (or the user’s perceived experience) without change to the existing data or methods. This is very interesting and smart (though perhaps slightly tricky). I am in favor.

**Strengths:**

+ Very interesting idea and very reasonable design.

+ Thorough experiments and good performance.

+ Good writing.

**Weaknesses:**

- I am very curious about the long motion sequence generation. As illustrated by authors, "By decomposing descriptions of long motions into a series of short motions using LLMs and subsequently concatenating these short motions into longer sequences, our Motion-Agent
can theoretically achieve infinite motion generation." Though I can generally understand the meaning, the details are unclear. Is "decomposition" done by GPT4 during long motion seq generation? If so, how it is achieved? Could authors provide detailed output of GPT4 of an example (like Figure 3)? Meanwhile, during training, will GPT4 be used to decompose HumanML3D data into shorter atomic units? If so, how does such labeling achieved and how to guarantee the alignment between motion and text?

- An important work is missed in related works as one of the first works that quantize/tokenize motion into GPT:
Bailando: 3d dance generation by actor-critic GPT with choreographic memory, CVPR 2022;
Bailando++: 3d dance GPT with choreographic memory, TPAMI 2023

**Questions:**

see above

---

> ### Author Response · Authors · 2024-11-22
> **Reply to Reviewer 7jDa**
>
> We thank you for your thoughtful reviews and suggestions.
>
> **GPT-4 for decomposition**
>
> When the prompt involves a request for a sequence of actions—such as "do A, then B, then C..."—Motion-Agent generates each individual short motion (e.g., A, B, C), which has been trained on the HumanML3D dataset. This dataset contains motion sequences of at most 10 seconds, covering a wide range of simple, "atomic" motions suitable for our purposes. During training, GPT-4 is not involved, as decomposition only occurs during inference.
>
> In long motion generation, GPT-4's role is analyzing and decomposing complex prompts into manageable segments. For example, in Figure 3, second turn, given the prompt, "Generate another motion where a person is kicked down and then stands up to fight back by slapping and kicking", the corresponding plan generated by GPT-4 is:
> 1. MotionLLM.generate('A person gets kicked down to the ground.')
> 2. MotionLLM.generate('A person pushes themselves up from the ground to stand up.')
> 3. MotionLLM.generate('A person slaps with their hand.')
> 4. MotionLLM.generate('A person delivers a kick.')
>
> Similarly, for some abstract prompts such as "floor exercise in artistic gymnastics" in Figure 3, GPT-4 interprets this prompt as a sequence of actions consisting of cartwheels, backflips, handstands, etc.
> By generating and concatenating these short motions, Motion-Agent effectively produces coherent, longer motion sequences.
>
> **Add citations**
>
> We appreciate the suggestion and we will include these papers in the revision.

---

> > ### Comment · Reviewer_7jDa · 2024-11-25
> >
> > I appreciate the authors' efforts in rebuttal. After reading authors' response, I make a clearer picture of this work.
> >
> > I got one more question on the coherence between GPT-4 and MotionLLM. As the authors said, the GPT-4 is not involved in training. In that case, will there be some gaps between GPT-4 and MotionLLM? How do you guarantee the instructions generated by GPT-4 can be all well performed in MotionLLM, e.g., the atomic motions are already well-learned by MotionLLM? Do you have any failure case to show? And what is the potential way to solve it.

---

> ### Author Response · Authors · 2024-11-25
>
> Thank you for your comments. We understand the concern about the potential gap between GPT-4 and MotionLLM. Since MotionLLM was originally pre-trained as an LLM model (i.e., GEMMA-2), it possesses strong language understanding capabilities. In our experiments, we observed that GPT-4 and MotionLLM communicate smoothly. As demonstrated, our Motion-Agent achieves state-of-the-art accuracy both quantitatively and qualitatively. We believe the issue you mentioned would be significant if MotionLLM had been trained from scratch.
>
> Additionally, we include few-shot examples in the instructional prompt given to GPT-4 to provide an overview of “simple and atomic” motion descriptions. As shown in the examples, rather than merely parsing user prompts like a small, lightweight LLM might, GPT-4 can generate plans with deeper reasoning. These outputs are more akin to expressions found in the HumanML3D dataset. To our surprise, GPT-4 can understand complex and abstract motions by explaining them in simple terms.
>
> Nevertheless, we agree that the HumanML3D dataset might not contain all types of atomic actions. Consequently, there is a possibility that an instruction from GPT-4 cannot be executed accurately by MotionLLM. We believe this issue can be resolved with the expansion of human motion datasets.

---

### Official Review · Reviewer_dbek · 2024-11-04

**Soundness:** 2
**Presentation:** 2
**Contribution:** 2
**Rating:** 6
**Confidence:** 4

**Summary:**

In this paper, the authors propose Motion-Agent, a conversational framework for human motion that utilizes large language models (LLMs). By incorporating MotionLLM, a generative agent fine-tuned with adapters, Motion-Agent enables bidirectional motion-text translation and supports multi-turn conversational interactions for tasks such as generation, captioning, and editing. The framework leverages the HumanML3D and KIT datasets to demonstrate its effectiveness, achieving competitive results by integrating GPT-4 for coordination without additional task-specific tuning.

**Strengths:**

1. The paper introduces Motion-Agent, a conversational framework that leverages pre-trained large language models (LLMs) for bidirectional human motion generation and understanding, achieving competitive results across various motion-language tasks.

2. The authors demonstrate Motion-Agent’s effectiveness through an extensive evaluation of the HumanML3D and KIT datasets, highlighting the model's competitive ability to generate and caption human motion sequences.

**Weaknesses:**

1. The proposed Motion-Agent framework seems to focus primarily on conversational motion generation, a capability that, according to the authors, could also be achieved using additional datasets for task-specific instruction tuning. While the authors assert that Motion-Agent is efficient, the experiments presented offer insufficient evidence to substantiate this claim.

2. The authors are encouraged to expand the discussion on the potential advantages of the proposed Motion-Agent framework. For instance, how does the model address out-of-domain motion concepts in comparison with current methods? A more thorough analysis of the model's generalization capabilities would contribute to a deeper understanding of its overall effectiveness.

3. The authors claim that Motion-Agent can theoretically achieve infinite motion generation. However, plots or tables illustrating changes with increasing conversation turns and motion lengths, should be provided to substantiate this claim.

4. The paper would benefit from ablation studies on the motion tokenizer, as well as comparisons with state-of-the-art RVQ-VAE models, such as MoMask which also employs RVQ-VAE to convert motion into a discrete representation.

5. The paper lacks comparisons on additional MotionGPT benchmarks, such as motion composition tasks. Tuning MotionLLM with these task is something that can be easily done.

**Questions:**

1. Did the authors attempt to fine-tune all parameters of the language models within MotionLLM? If so, what were the resulting outcomes, and how did they compare to the model tuned with LoRA?

2. Is it feasible to extend the Motion-Agent framework to incorporate vision and audio modalities, given that GPT-4o is known to support multimodal inputs? What impact might this integration have on the model's performance and potential applications?

---

> ### Author Response · Authors · 2024-11-22
> **Reply to Reviewer dbek (part1/2)**
>
> We thank you for your thoughtful reviews and suggestions.
>
> **Efficiency of Motion-Agent**
>
> The Motion-Agent framework focuses on conversational motion generation by combining MotionLLM for motion-language translation and GPT-4 for conversational control. Instead of building other task-specific datasets that restrict to a finite set of problems using LLMs, we use LLMs for such tasks without dataset collection and training, and hence more flexible and efficient.
>
> For example, MotionChain (Jiang et al., 2024) depends on manually constructed datasets. A single-turn conversation dataset (95k training entries and 18k testing entries) is created by sampling HumanML3D with GPT-4 to form prompt-response pairs. However, this dataset is limited to a narrow range of tasks and queries, restricting its adaptability to out-of-domain scenarios. For multi-turn dialogues, MotionChain samples a large multi-turn dataset from the single-turn dataset resulting in inefficiencies. Instead, Motion-Agent leverages existing motion-language datasets such as HumanML3D for translation tasks, while employing GPT-4’s generalization capabilities for conversational control. We believe our approach can better leverage existing data, and thus significantly enhance efficiency and scalability.
>
> Moreover, Motion-Agent is training-efficient for generating longer motions. For example, instead of training on all permutations of short motions, MotionLLM is trained to generate each motion independently, allowing them to be combined as needed and appropriate. This design avoids unnecessary computational overhead and enhances the framework’s adaptability and scalability.
>
> **Generalization capabilities of Motion-Agent**
>
> MotionLLM is trained on a dataset containing approximately 20K motions, covering the majority of simple and atomic human motions. When faced with complex, abstract out-of-domain prompts, GPT-4 can decompose them into simpler, in-domain motions, so that Motion-Agent can generate the motion by producing each component individually and then smoothly concatenating them. In contrast, other models remain constrained to the domain of their training data.
>
> **Generation length**
>
> In practice, the length of motion generation is constrained by several factors, including the context length of the conversational LLM (e.g., GPT-4) and hardware limitations, such as memory and processing power. However, theoretically, the decoder itself has no restrictions on the motion length and can ensure smooth transitions between motions. Since MotionLLM can be invoked repeatedly, it is possible to generate motion sequences of virtually unlimited length. This means that, in theory, Motion-Agent can achieve infinite motion generation by repeatedly invoking MotionLLM to generate motion tokens and concatenating them into longer sequences.
>
> **Comparisons with state-of-the-art RVQ-VAE models**
>
> We provide an additional ablation study, please refer to the general response.
>
> **Additional MotionGPT benchmarks**
>
> MotionLLM does not explicitly address motion composition tasks, as these are primarily handled by the planner agent, GPT-4. The planner generates intermediate descriptions of motion, which are then interpreted by MotionLLM to produce the final composition. While MotionLLM can indeed be fine-tuned on additional tasks, such as motion prediction and in-between generation (as demonstrated below), and has shown competitive performance in these areas, our work does not focus on such tasks. We believe that motion composition is better suited to the planner agent, as this approach provides greater flexibility and supports more intuitive conversational editing. MotionLLM’s core design is focused on translational tasks, where it has already demonstrated strong performance in both motion generation and captioning.
>
> |        | Motion Prediction |                |             |             | Motion In-between |                 |            |
> |---------------|-------------------|----------------|-------------|-------------|-------------------|----------------|-------------|
> |  **Methods**  | FID&nbsp;↓        | Diversity&nbsp;↑ | ADE&nbsp;↓  | FDE&nbsp;↓  | FID&nbsp;↓        | Diversity&nbsp;↑ | ADE&nbsp;↓  |
> | MDM       | 6.031             | 7.813          | 5.446       | 8.561       | 2.698             | 8.420          | 3.787       |
> | MotionGPT | 0.905             | 8.972          | 4.745       | 6.040       | **0.214**         | 9.560          | **3.762**   |
> | MotionLLM | **0.716**         | **9.836**      | **4.641**   | **5.057**   | 0.286             | **9.678**      | 3.789       |
>
> *Comparison of motion prediction and motion in-between on part of AMASS dataset.*
>
> **LoRA vs Full fine-tune**
>
> Please refer to the general response.

---

> ### Author Response · Authors · 2024-11-22
> **Reply to Reviewer dbek (part2/2)**
>
> **More modalities**
>
> Indeed, GPT-4o is renowned for its multimodal capabilities, and we believe it is feasible to integrate visual and auditory inputs into our framework, potentially enhancing both performance and application scope.
>
> In our initial experiments, we have observed promising results:
>
> Visual Input: When provided with an image of a volleyball player along with a prompt to generate a sequence of motions corresponding to the picture, Motion-Agent successfully produces a series of actions such as receiving and spiking. This demonstrates the framework's ability to interpret visual cues and translate them into meaningful motion plans.
>
> Audio Input: Similarly, when given an audio description of the desired motion, GPT-4o accurately transcribes the voice input into textual prompts. Motion-Agent then generates detailed plans by invoking MotionLLM based on these prompts. This showcases the potential for auditory information to guide motion generation.
>
> Integrating vision and audio modalities can significantly enhance the model's performance by providing richer contextual information, leading to more accurate and diverse motion outputs. It also opens up new avenues for applications in areas like virtual reality, human-computer interaction, and animation, where multimodal inputs are valuable.
>
> We recognize the great potential in incorporating additional modalities and are committed to exploring this direction in our future work.

---

> > ### Comment · Reviewer_oMaw · 2024-11-27
> > **Wish to see your feedback.**
> >
> > Dear reviewer colleague,
> > Can you read this rebuttal and comment on whether it helps resolve your concerns?
> > Best

---

### Comment · Reviewer_oMaw · 2024-11-21
**No author responses?**

Hi, all.
I received an email to check the author's responses, but I cannot find any of this paper. Can anyone confirm that authors did not submit them?
Best

---

### Author Response · Authors · 2024-11-22
**General Response**

We thank all reviewers for their insightful comments. We refer to dbek, 7jDa, DEqA, QaCH, and oMaw as R1, R2, R3, R4 and R5, respectively.

We would like to first reiterate some key aspects of Motion-Agent that may address some of the reviewers' questions. The primary goal of Motion-Agent is to efficiently handle complex and flexible motion-language tasks through multi-turn conversations using LLMs. LLMs such as GPT-4 do not generate or understand human motions, so we employ a translation agent, which is our *MotionLLM*, to bridge this gap. MotionLLM enables bi-directional generation, which can support the Motion-Agent framework for the generation and understanding tasks. Among existing models capable of both motion generation and motion captioning, MotionLLM delivers state-of-the-art performance.
MotionLLM is built upon a lightweight LLM and fine-tuned using LoRA, thus avoiding the need for pre-training from scratch. It is then integrated with another LLM (e.g., GPT-4) for conversational tasks, without requiring additional training. Then GPT-4 and MotionLLM work hand-in-hand, where GPT-4 interprets user requests and decomposes them into plans consisting of simple calls to MotionLLM. The generated motions can be concatenated to produce significantly longer sequences. By breaking down complex tasks, such as editing, composition, or question answering, into manageable components for MotionLLM, our framework achieves remarkable efficiency without requiring task-specific data or further training.

In the following, we provide additional experiments requested by reviewers.

**Ablation study on different tokenizers (R1, R3, R5)**

We conducted additional experiments replacing our current VQ-VAE tokenizer with the RVQ-VAE tokenizer used in MoMask. RVQ-VAE differs from traditional VQ-VAE by using $Q+1$ ordered codebooks. The first codebook is used for base tokens, similar to traditional VQ, while the remaining $Q$ codebooks are used to represent residuals for enhancing fidelity.

Following MoMask's approach, the MotionLLM in our case is aware of the base layer tokens. To predict the residual layer token sequences, we follow MoMask to use an additional non-autoregressive (NAR) transformer that takes as input the base layer token sequences from the LLM output. This NAR transformer can thus be considered as part of our detokenizer, which does not affect the LLM’s inference or training process.

The results are shown as follows:
|Models|Trainable Params|Top 1 R Precision ↑|Top 3 R Precision ↑| FID ↓| Multimodal Dist ↓| Diversity ↑|
|-|-|-|-|-|-|-|
|Gemma2-2b R=32 VQ|41.5M|0.415|0.750|0.712|2.938|11.251|
|Gemma2-2b R=64 VQ|83.1M|0.422|0.762               | 0.658  | 2.929             | 11.195      |
| Gemma2-9b R=32 VQ          | 108M             | 0.439               | 0.776               | 0.438  | 2.872             | 11.151      |
| Gemma2-2b R=64 RVQ         | 83.1M + 13.4M    | 0.429               | 0.768               | 0.647  | 2.857             | 10.126      |
| Gemma2-2b Full VQ          | 2697.4M          | 0.423               | 0.774               | 0.591  | 2.913             | 11.138      |
*More Comparisons on KIT-ML.*

Our results show that using RVQ-VAE does not significantly improve performance compared to the original VQ-VAE, validating that the VQ-VAE model is already sufficiently effective for our tasks. While RVQ-VAE provides some potential improvements in fidelity, it introduces additional training overhead (the 13.4M parameters are from the NAR transformer). Specifically, the NAR transformer required for residual token prediction is challenging to train and adds complexity to the system (namely, requiring an additional training step for the NAR transformer).

Notwithstanding, we believe that both VQ-VAE and RVQ-VAE are effective for motion tokenization, and that the choice between them is more of an engineering decision based on the specific trade-offs involved. Our Motion-Agent framework remains flexible and can support different tokenizers, as long as they can convert motions into discrete representations. Future advancements in tokenization methods may further enhance the framework's performance and capabilities.

**Ablation study on LoRA vs Full fine-tuning (R1)**

We also experimented fine-tuning all the parameters of LLM, the corresponding result is also shown in the table above. When using a 2b backbone, full fine-tuning does improve overall scores but at the costly expense of significantly increasing training overhead, which is in stark contrast to  LoRA fine-tuning a 9b backbone model which requires over 20 times fewer parameters trained. Fully training such a large model also demands substantially more computational power and memory, making it nearly impossible on consumer-level GPUs. This result indicates that LoRA is indeed a more efficient and effective approach than full fine-tuning, suggesting it is a better alternative for scaling up to a larger backbone in the future.

---

### Author Response · Authors · 2024-11-27
**Revision**

We thank the reviewers and have addressed their suggestions in the revision by:

- Adding citations for Bailando and Bailando++.
- Including the additional ablation studies.

---

### Meta-Review · Area_Chair_mRHw · 2024-12-20

**Metareview:**

The submission introduces a framework for human motion modeling based on conversations.  Reviewers were in general positive about the submission, appreciating the novel idea and the good writing.  The rebuttal was also helpful in convincing the reviewers to level up their support.  Reviewer DEqA was less positive, though their review is low-quality, and they did not engage in the discussion.  The AC agreed with the majority and recommended acceptance.

**Additional Comments On Reviewer Discussion:**

Reviewer DEqA did not engage in the submission, while all other reviewers were positive.

---

### Decision · Program_Chairs · 2025-01-22

Accept (Poster)